# Adjusting for Autocorrelated Errors in Neural Networks for Time Series

**Fan-Keng Sun**
Department of EECS, MIT
fankeng@mit.edu

**Christopher I. Lang**
Department of EECS, MIT
langc@mit.edu

**Duane S. Boning**
Department of EECS, MIT
boning@mtl.mit.edu

## Abstract

An increasing body of research focuses on using neural networks to model time series. A common assumption in training neural networks via maximum likelihood estimation on time series is that the errors across time steps are uncorrelated. However, errors are actually autocorrelated in many cases due to the temporality of the data, which makes such maximum likelihood estimations inaccurate. In this paper, in order to adjust for autocorrelated errors, we propose to learn the autocorrelation coefficient jointly with the model parameters. In our experiments, we verify the effectiveness of our approach on time series forecasting. Results across a wide range of real-world datasets with various state-of-the-art models show that our method enhances performance in almost all cases. Based on these results, we suggest empirical critical values to determine the severity of autocorrelated errors. We also analyze several aspects of our method to demonstrate its advantages. Finally, other time series tasks are also considered to validate that our method is not restricted to only forecasting.

## 1 Introduction

Time series data are increasingly ubiquitous as the cost of data collection continues to decrease. Analysts seek to model such data even when the underlying data-generating process (DGP) is unknown, in order to perform various tasks such as time series forecasting, classification, regression, and anomaly detection. For example, industrial internet of things (IIoT) sensors are being installed around manufacturing equipment to collect time series data that can help to increase the efficiency of day-to-day manufacturing operations.

During the collection and modeling of time series, there are inevitably errors. It is common to assume that errors at different time steps are uncorrelated, especially for fitting neural networks (NNs) under the framework of maximum likelihood estimation (MLE). However, errors are actually oftentimes *autocorrelated* due to the temporality of the data interacting with the three sources noted below.

The first source is the omission of influential variables. In real-world situations, determining which variables to collect and include in the model is a complicated task. For example, temperature should be an important variable to consider if we want to forecast household electricity consumption. But publicly available electricity datasets, which many previous works are compared upon [37, 38, 54, 57, 64], do not include such information. Even if we were to add temperature as a variable, there are still many other variables that might be influential. In many real-world cases, in order to have satisfactory prediction results, it is either too complicated to decide which and how many variables are required, or impossible to collect all the required variables. Omitting influential variables can result in autocorrelated errors.

Secondly, measurement errors are almost unavoidable, and measurement errors in time series are usually autocorrelated due to the temporality of the data. For example, when taking time-sampled measurements in semiconductor fabrication equipment, sensor noise may exist due to drift, calibration

35th Conference on Neural Information Processing Systems (NeurIPS 2021).

error, or environmental factors. If the rate of measurement is faster than the rate of fluctuation of the noise, then the errors are autocorrelated.

The third important source is model misspecification. In general, it is difficult to formulate an exact model for the underlying DGP. Such model misspecification can cause errors to be autocorrelated even if an optimal fit within the limitation of the approximate model is reached. A simple example would be fitting a linear time-trend model on a sinusoidal signal. Neural networks (NNs), as universal approximators, help alleviate this issue and enable us to achieve good approximations in many cases. However, even when an optimal fit is reached for a given NN structure, this does not imply that the remaining errors are uncorrelated.

All three aforementioned sources can lead to *autocorrelated errors* [26, 46], in which errors at the current time step are correlated with errors at previous time steps.

In ordinary least squares (OLS), autocorrelated errors violate the assumption that the errors are uncorrelated, which implies that the Gauss-Markov theorem is not applicable. Specifically, the variance of the coefficient estimates increases but the estimated standard error is underestimated. Thus, if neglected, prediction accuracy is reduced, and an outcome that appears to be statistically significant may actually be insignificant. As for nonlinear data, autocorrelated errors impair the standard MLE and thus weaken model performance.

Adjusting for autocorrelated errors in linear or nonlinear time series data has been studied extensively, especially in econometrics [4, 13, 20, 21, 24, 29, 49]. However, those methods are applicable only when the exact (and correct) form of the underlying system is known. On the other hand, NNs for time-series-related tasks [5, 10, 17, 22] have become a popular research direction due to NNs' effectiveness in approximating unknown, nonlinear systems. However, to the best of our knowledge, none of the existing NN-based methods adjust for autocorrelated errors. In this work, we introduce a method to account for autocorrelated errors in NNs and show that the proposed method improves the performance in many time series tasks, especially time series forecasting. The implementation of our method can be found at https://github.com/Daikon-Sun/AdjustAutocorrelation.

Our main contributions are:

- We propose to learn the autocorrelation coefficient jointly with model parameters via gradient descent in order to adjust for autocorrelated errors in NNs for time series.

- Our large-scale experiments on time series forecasting show that our method improves performances across a wide range of real-world datasets and state-of-the-art NN architectures.

- Based on these large-scale experiments, we identify the issue of autocorrelated errors and we suggest empirical critical values of the remaining autocorrelation in errors that act as a guideline to determine whether adjusting for autocorrelated errors is necessary.

- By ablation study and grid-searching over autocorrelation coefficients and model hyperparameters, we validate the strength of our method. We also study the effect of model misspecification.

- We apply our methods on other time series tasks to demonstrate the effectiveness of our method on a variety of additional time series tasks beyond forecasting.

## 2 Preliminaries

### 2.1 Time series forecasting

In time series forecasting, we have an input matrix $\mathbf{X} = \{\mathbf{X}_1, \ldots, \mathbf{X}_t, \ldots, \mathbf{X}_T\} \in \mathbb{R}^{T \times N}$ representing $N$ variables sampled at the same rate at the same time for $T$ time steps where $\mathbf{X}_t \in \mathbb{R}^N$ is the $t$-th sample. The goal is to forecast the value of $\mathbf{X}_t$ given the histories $\{\mathbf{X}_1, \ldots, \mathbf{X}_{t-1}\}$. In practice, only the $W$ most recent histories $\{\mathbf{X}_{t-W}, \ldots, \mathbf{X}_{t-1}\}$ are fed into a model. This is a common approach [2, 37, 50, 57] that assumes older histories are less informative, establishes fair comparisons between different methods, and makes the memory usage plausible.

Mathematically speaking, given the model $f$, we optimize the model parameters $\theta$ to minimize the mean squared error (MSE):

$$\text{MSE} = \sum_t \|e_t\|_2^2 = \sum_t \|\mathbf{X}_t - \hat{\mathbf{X}}_t\|_2^2, \tag{1}$$

where $\hat{\mathbf{X}}_t = f(\mathbf{X}_{t-1}, \ldots, \mathbf{X}_{t-W}; \theta)$ is the model forecast, $\mathbf{X}_t = \hat{\mathbf{X}}_t + e_t$, and $e_t$ is the error.

## 2.2 Autocorrelated Errors

In most machine learning literature and in our work, the errors are assumed to be uncorrelated across different series. Thus, for ease of understanding, we look at each series separately and assume $e_t \in \mathbb{R}$.

Usually, the errors $e_t$ are assumed to be independent, identical, and normally distributed:

$$\text{Cov}(e_{t-\Delta_t}, e_t) = 0, \ \forall \Delta_t \neq 0, \tag{2}$$

$$e_t \sim \mathcal{N}(0, \sigma^2). \tag{3}$$

Thus, minimizing MSE is equivalent to maximizing likelihood. However, as discussed in Section 1, there are often situations in which the assumption may be violated and the errors are thus autocorrelated. In general, a $p$-th order autocorrelated error has the form

$$e_t = \rho_1 e_{t-1} + \cdots + \rho_p e_{t-p} + \epsilon_t, \ |\rho_i| < 1, \forall i \tag{4}$$

where $\rho_1, \ldots, \rho_p$ are autocorrelation coefficients and $\epsilon_t \sim \mathcal{N}(0, \sigma^2)$ is the uncorrelated error. Notice that the magnitude of $\rho_i$ should be strictly smaller than 1, as explained in Appendix A.

Generally, the first-order autocorrelation is the single most significant term because it is reasonable to assume that the correlation between $e_t$ and $e_{t-\Delta_t}$ decreases when $\Delta_t$ increases. Thus, in this work, we only focus on the linear, first-order autocorrelation following previous work [4, 13, 16, 29, 49] and simplify the notation to $e_t = \rho e_{t-1} + \epsilon_t$. Nevertheless, our method can be extended to higher order in cases where other terms are important.

It is possible that a nonlinear, more complex structure would be beneficial, but we didn't discuss it here because (1) a linear model is a good choice for simple and basic modeling, (2) we follow the work in the field of econometrics, where most of the discussions about autocorrelated errors happen, and most importantly (3) if we use a NN for modeling the errors (i.e., replacing $\mathbf{X}_t - \rho\mathbf{X}_{t-1}$ with $\mathbf{X}_t - f(\mathbf{X}_{t-1}; \theta)$), the overall model becomes deeper with many more parameters, then it becomes unclear whether the improvement comes from adjustment or from a bigger model.

When there exists first-order autocorrelated errors, as derived in Appendix A,

$$\text{Cov}(e_t, e_{t-\Delta_t}) = \frac{\rho^{\Delta_t}}{1 - \rho^2}\sigma^2, \forall \Delta_t = 0, 1, 2, \ldots, \tag{5}$$

i.e., errors are no longer correlated, and thus the standard MLE becomes untenable. Alternatively, one should use the following form of MLE:

$$\mathbf{X}_t - \rho\mathbf{X}_{t-1} = f(\mathbf{X}_{t-1}, \ldots, \mathbf{X}_{t-W}; \theta) - \rho f(\mathbf{X}_{t-2}, \ldots, \mathbf{X}_{t-W-1}; \theta) + \epsilon_t, \tag{6}$$

so that the remaining errors are uncorrelated. Practically, the true $\rho$ value is unknown, so an estimate $\hat{\rho}$ is used instead. Per Section 3.1, there are several methods to obtain the estimate $\hat{\rho}$ with linear or predetermined nonlinear models, but none for NNs.

## 3 Related Work

### 3.1 Adjusting for Autocorrelated Errors

Adjusting for autocorrelated errors in linear models has been studied extensively, especially in econometrics. Typically, after collecting the data and determining the formulation of the underlying system, the Durbin-Watson statistic [16] is calculated to detect the presence of first-order autocorrelated errors. If there is statistical evidence that first-order autocorrelated errors exist, then one of the following methods can be applied.

The Cochrane-Orcutt method [13] is the most basic approach. It first estimates the remaining autocorrelation in the errors, then transforms the series to weaken the autocorrelation and fits OLS to the transformed series. The procedure can be done once or iteratively until convergence; both versions have the same asymptotic behavior [21], but the performance may differ with finite samples. During the transformation of the time series, the first sample is discarded — a large information loss when the sample size is small. The Prais-Winsten method [49] solves this issue by retaining

the first sample with appropriate scaling. Finally, the Beach-Mackinnon method [4] formulates the exact likelihood function that incorporates not only the first sample but also an additional term that constrains the autocorrelation coefficient to be stationary.

These methods update the autocorrelation coefficient starting from $0$. However, multiple local minima might exist and the procedure might converge to a bad local minimum [14, 15, 48, 55]. Thus, the Hildreth-Lu method [29] grid-searches over autocorrelation coefficients and picks the best one.

For autocorrelation of higher orders, the Ljung-Box test [39] or Breusch-Godfrey test [9, 25] can be applied to detect their presence. Aforementioned methods can then be extended for adjustment [1, 8].

When dealing with nonlinear data, as long as the underlying system is known, prior methods are applicable by changing OLS to nonlinear least squares [21] or employing other nonlinear optimization techniques such as BFGS [18]. However, it is often difficult to assume or formulate a correct and exact model structure for real-world time series data, especially when there are multiple series. This is where NNs come into play.

Neural networks (NNs) [31] are inherently designed to learn arbitrary nonlinear relationships between input-target pairs from the data. In [32], it has been proven that a nonlinear NN with sufficient number of hidden units is capable of approximating any function to any degree of accuracy. Pairing this with the exponentially increasing availability of data, computational power, and new algorithms, NNs can learn complex functionalities. Although NNs show promising outcomes, as of today, there is no publication we are aware of that is dedicated to adjusting autocorrelated errors in NNs for time series.

### 3.2 Time series forecasting

For modeling linear and univariate time series, the autoregressive integrated moving average (ARIMA) [8] is the most well-known among autoregression (AR) models. Vector autoregression (VAR) generalizes AR models to multivariate time series, but still only for linear data. For nonlinear data, previous methods, including kernel methods [11], ensembles [7], Gaussian processes [19], and regime switching [58] apply predetermined nonlinearities which may fail to capture the complex nonlinearities in real-world datasets. Thus, NNs have become prominent for time series forecasting. The four popular building blocks for NNs are recurrent neural networks (RNNs), convolutional neural networks (CNNs), graph neural networks (GNNs), and Transformers.

Long Short-term Memory (LSTM) [30] networks are the most basic NNs for time series forecasting. These are an improved version of RNNs, but these often fail at learning complex temporal (intra-series) and spatial (inter-series) patterns. Temporal Pattern Attention [57] networks add convolutional attention to LSTMs to help capture temporal patterns. Adaptive Graph Convolutional Recurrent Networks [2] combine RNNs and GNNs to learn not only temporal but also spatial patterns. The Temporal Convolutional Networks [3] are dilated CNNs with residual connections that are good at modeling long sequences. The Convolutional Transformer [38] plugs an input convolution into the original Transformer [60] to capture local temporal patterns. Dual Self-Attention Networks [33] use CNNs for temporal patterns and self-attention for spatial patterns.

There are many other works using NN-based models for time series forecasting [37, 50, 54, 56, 63], most of which include at least one of the four building blocks above. However, to the best of our knowledge, none of the previous work addresses the issue of autocorrelated errors.

## 4 Our Method

Following the work of [13, 21, 49], it is straightforward to design a naive procedure to adjust for autocorrelated errors in NNs:

1. Initialize model parameter $\theta$ randomly and $\hat{\rho}$, the estimate of $\rho$, at $0$.

2. Fix $\hat{\rho}$ and train the model sufficiently to minimize MSE on the training data:

$$\mathbf{X}_t - \hat{\rho}\mathbf{X}_{t-1} = f(\mathbf{X}_{t-1}, \ldots, \mathbf{X}_{t-W}; \theta) - \hat{\rho}f(\mathbf{X}_{t-2}, \ldots, \mathbf{X}_{t-W-1}; \theta) + e_t \qquad (7)$$

and obtain the new model parameter $\theta'$.

3. Compute the errors $e_t = \mathbf{X}_t - f(\mathbf{X}_{t-1}, \ldots, \mathbf{X}_{t-W}; \theta')$.

Table 1: RRMSE and average relative improvement of all combinations of models and datasets averaged over five runs. "w/o" implies without adjustment for autocorrelated errors, whereas "w/" implies with adjustment. Best performance is in boldface and is superscribed with † if the p-value of paired t-test is lower than $5\%$. Average relative improvement is the percentage of improvement of "w/" over "w/o" averaged over all datasets for each model.

| Models | LSTM | | TPA | | AGCRN | | TCN | | Conv-T | | DSANet | |
|---|---|---|---|---|---|---|---|---|---|---|---|---|
| Datasets | w/o | w/ | w/o | w/ | w/o | w/ | w/o | w/ | w/o | w/ | w/o | w/ |
| PeMSD4 | .2304 | **.1960**† | .1742 | **.1737** | .1718 | **.1709** | .2203 | **.1965**† | .2111 | **.2055**† | .1775 | **.1762** |
| PeMSD8 | .1960 | **.1586**† | .1392 | **.1388** | .1370 | **.1366** | .1809 | **.1587**† | .1679 | **.1516**† | .1409 | **.1398** |
| Traffic | .4936 | **.3643**† | .3559 | **.3517**† | **.3326**† | .3339 | .4657 | **.3678**† | .4645 | **.3711**† | .3454 | **.3444** |
| ADI-920 | .0469 | **.0432**† | .0436 | **.0419**† | .0491 | **.0474**† | .0438 | **.0421**† | .0681 | **.0561**† | .0992 | **.0599**† |
| ADI-945 | .0060 | **.0060** | .0545 | **.0368**† | .0095 | **.0084**† | .0065 | **.0064** | .0358 | **.0355** | .0709 | **.0581** |
| M4-Hourly | .0444 | **.0328**† | .0520 | **.0416**† | **.0282** | .0284 | .0358 | **.0317**† | .0422 | **.0414** | .0902 | **.0773**† |
| M4-Daily | .8886 | **.0429**† | .0485 | **.0321**† | .0334 | **.0311**† | .6802 | **.0933**† | .6961 | **.2942**† | .0309 | **.0302** |
| M4-Weekly | 1.241 | **.5826**† | .2247 | **.1758** | .5406 | **.4431**† | 1.065 | **.4958**† | 1.214 | **.9989**† | .0412 | **.0396**† |
| M4-Monthly | .9112 | **.3868**† | .2777 | **.1564**† | .1540 | **.1308**† | .5715 | **.3601**† | .8964 | **.6939**† | .1171 | **.1096**† |
| M4-Quarterly | .6536 | **.4115**† | .2779 | **.2067**† | .1776 | **.1637**† | .4946 | **.4184**† | .5207 | **.4679**† | .1624 | **.1439**† |
| M4-Yearly | .6177 | **.4207**† | .4024 | **.2662** | .3241 | **.2446**† | .5910 | **.4112**† | .4657 | **.4203** | .1319 | **.1135** |
| M5-L9 | .2760 | **.2260**† | .1898 | **.1854** | .1959 | **.1940** | .2754 | **.2260**† | .2890 | **.2616**† | **.1823** | .1824 |
| M5-L10 | .6035 | **.3787**† | .3227 | **.3155** | .3029 | **.2999** | .5577 | **.4471**† | .5603 | **.4127**† | .3066 | **.3043**† |
| Air-quality | .2014 | **.1764**† | .1715 | **.1677**† | .1700 | **.1696** | .1926 | **.1787**† | .1934 | **.1706**† | .1695 | **.1687** |
| Electricity | .0840 | **.0793**† | .0648 | **.0639**† | .0759 | **.0719**† | .0799 | **.0730**† | .0790 | **.0741**† | **.0663** | .0664 |
| Exchange | .0815 | **.0188**† | .0509 | **.0354** | .0124 | **.0119** | .0822 | **.0266**† | .0394 | **.0366** | .0115 | **.0109**† |
| Solar | .1517 | **.1055**† | .1061 | **.1052**† | .0994 | **.0992** | .1407 | **.1057**† | .1389 | **.1071**† | .1064 | **.1032**† |
| Avg. rel. impr. | | 32.4% | | 15.1% | | 5.79% | | 25.3% | | 15.0% | | 7.12% |

4. Use the errors to update $\hat{\rho}$ by linearly regressing $e_t$ on $e_{t-1}$, i.e.,

$$\hat{\rho} = \frac{\sum_{t=2}^{T} e_t e_{t-1}}{\sum_{t=1}^{T-1} e_t^2}. \tag{8}$$

5. Go back to step 2 or stop if sufficiently converged.

Empirically, we find that we cannot successfully train NNs by following the naive procedure. Thus, we identify several issues in the naive procedure and propose solutions to address those issues.

First, in the naive procedure, $\hat{\rho}$ and $\theta$ are not optimized jointly. Instead, $\hat{\rho}$ and $\theta$ are optimized alternatively, which can be considered as coordinate descent [62]. This makes $\hat{\rho}$ prone to converge to bad local minimum [14, 15, 48, 55] on even simple, linear data. Instead of coordinate descent, we propose to treat $\hat{\rho}$ as a *trainable* parameter and update it with $\theta$ jointly using stochastic gradient descent (SGD). Using SGD has the benefit of escaping bad local minima.

The second issue is that the target $\mathbf{X}_t - \hat{\rho}\mathbf{X}_{t-1}$ is not directly related to the model output. Instead, the target is related to the difference of two model outputs, which complicates the optimization. Hence, we approximate the right-hand-side of Equation (7) with just one model for the same set of inputs while treating $\hat{\rho}$ as a trainable parameter:

$$f(\mathbf{X}_{t-1}, \ldots, \mathbf{X}_{t-W}; \theta) - \hat{\rho} f(\mathbf{X}_{t-2}, \ldots, \mathbf{X}_{t-W-1}; \theta) \simeq f(\mathbf{X}_{t-1}, \ldots, \mathbf{X}_{t-W-1}; \theta, \hat{\rho}). \tag{9}$$

Now, the minimization of MSE becomes

$$\mathbf{X}_t - \hat{\rho}\mathbf{X}_{t-1} = f(\mathbf{X}_{t-1}, \ldots, \mathbf{X}_{t-W-1}; \theta, \hat{\rho}). \tag{10}$$

This simplifies the optimization by merging two model outputs into one.

Next, per Equation (10), the input series and target series are not in the same form. Namely, the input series is in the form of $\mathbf{X}_t$ whereas the target series is in the form of $\mathbf{X}_t - \hat{\rho}\mathbf{X}_{t-1}$. Usually, in time series forecasting, the input series and target series have the same form. Thus, we modify Equation (10) into

$$\mathbf{X}_t - \hat{\rho}\mathbf{X}_{t-1} = f(\mathbf{X}_{t-1} - \hat{\rho}\mathbf{X}_{t-2}, \ldots, \mathbf{X}_{t-W} - \hat{\rho}\mathbf{X}_{t-W-1}; \theta), \tag{11}$$

so both the input and target series are in the form of $\mathbf{X}_t - \hat{\rho}\mathbf{X}_{t-1}$.

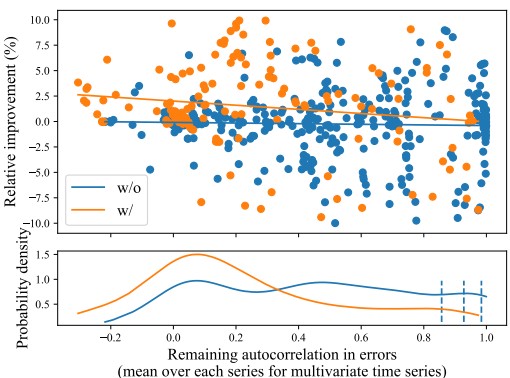

Figure 1: Scatter plot of all runs in Section 5.4 and its density plot in the $x$-dimension. "w/o" implies without adjustment for autocorrelated errors, whereas "w/" implies with adjustment. Runs that have a larger than 10% relative improvement are considered as outliers and thus excluded. The solid lines in the scatter plot are regression lines of the corresponding methods and the dashed lines in the density plot are the critical values for 10%, 5%, and 1% right-tailed probabilities of the w/o method.

Additionally, we would like to constrain $|\hat{\rho}| < 1$ as in Equation (4); in practice, we employ the transformation $\hat{\rho} := \tanh(\hat{\rho})$.

Finally, since we can only use exactly $W$ histories as described in Section 2.1, we replace $\mathbf{X}_{t-W-1}$ in Equation (11) by the mean vector $\bar{\mathbf{X}} \in \mathbb{R}^N$ in the training set.

After all these modifications, we can now successfully train NNs for time series forecasting using SGD just like the standard MLE in previous works [2, 33, 37, 38, 57, 63] while adjusting for autocorrelated errors. Notice that if $\hat{\rho} = 0$, our formulation reduces exactly to the standard MLE. Also, our method is applicable to non-Gaussian errors as long as all errors have the same probability distribution and follow Equation (4); just that instead of MSE, a different loss function would be used.

# 5 Experiments on Time Series Forecasting

## 5.1 Models

To demonstrate that our method is model-agnostic, we apply it on six different models, including basic and state-of-the-art models: (1) Long short-term memory (LSTM) [30]; (2) Temporal Pattern Attention (TPA) [57]; (3) Adaptive Graph Convolutional Recurrent Network (AGCRN) [2]; (4) Temporal Convolutional Network (TCN) [3]; (5) Convolutional Transformer (Conv-T) [38]; and (6) Dual Self-Attention Network (DSANet) [33]. These six models are chosen deliberately to include RNN, CNN, GNN, and Transformer, the four building blocks of NNs for time series forecasting.

We do not include univariate models, such as DeepAR [54] and N-BEATS [47] because we focus on multivariate forecasting with a single NN-based model.

## 5.2 Datasets

A wide range of datasets are explored to show that the benefit of applying our method is not dependent on the dataset. We categorize these datasets into five groups:

- Traffic (PeMSD4, PeMSD8, Traffic) [27, 37]: road occupancy rate. PeMSD4 and PeMSD8 are used in AGCRN. Traffic is used in both TPA and Conv-T.

- Manufacturing (ADI-920, ADI-945): sensor values during semiconductor manufacturing.

- M4-competition (M4-Yearly, M4-Quarterly, M4-Monthly, M4-Weekly, M4-Daily, M4-Hourly) [42]: re-arranged data from the M4-competition where each dataset consists of series from microeconomics, macroeconomics, financial, industry, demographic, and other. M4-Hourly is used in Conv-T.

Table 2: RRMSE for the best grid-searched hyperparameters averaged over ten runs. Best performance in boldface and is superscribed with † if the p-value of paired t-test is lower than $5\%$.

| Datasets | Models | w/o | w/ |
|---|---|---|---|
| M4-Hourly | AGCRN | .0275 | **.0267**† |
| Traffic | AGCRN | **.3325**† | .3332 |
| M5-L9 | DSANet | .1765 | **.1763** |
| Electricity | DSANet | .0651 | **.0641**† |

- M5-competition (M5-L9, M5-L10) [43]: aggregated Walmart data from the M5-competition.

- Miscellaneous (Air-quality, Electricity, Exchange-rate, Solar) [37]: other datasets. Electricity and Solar are used in both Conv-T and TPA. Exchange-rate is used in TPA.

Each dataset is split into training (60%), validation (20%), and testing (20%) set in chronological order following [2, 37, 50, 57]. We do not follow the $K$-fold cross-validation (CV) proposed in [6] because $K$-fold CV is too costly for a single run. In data preprocessing, the data is normalized by the mean and variance of the whole training set. The detailed description and statistics of the datasets can be found in Appendix B.

### 5.3 Comparison metrics

Our first metric is the root relative mean squared error (RRMSE) [2, 37, 50, 57] on the testing set:

$$\text{RRMSE} = \frac{\sqrt{\sum_{t \in \text{testing}} \|\mathbf{X}_t - \hat{\mathbf{X}}_t\|_2^2}}{\sqrt{\sum_{t \in \text{testing}} \|\mathbf{X}_t - \bar{\mathbf{X}}\|_2^2}}, \tag{12}$$

where $\bar{\mathbf{X}}$ is the mean value of the whole testing set. The benefit of using RRMSE is to scale the errors so the outcomes are more readable, regardless of the scale of the dataset. To aggregate results over multiple datasets, our second metric is the averaged relative improvement:

$$\frac{1}{D} \sum_{d=1}^{D} \frac{(\overline{\text{RRMSE}}_{\text{w/o, d}} - \overline{\text{RRMSE}}_{\text{w/, d}})}{\overline{\text{RRMSE}}_{\text{w/o, d}}} \cdot 100\%, \tag{13}$$

where $D$ is the number of datasets, w/o denotes training without adjustment, w/ denotes with adjustment, and $\overline{\text{RRMSE}}$ is the averaged RRMSE over multiple runs.

### 5.4 Evaluation on all combinations

We run all combinations of models and datasets with and without our method of adjusting for autocorrelated errors. Due to computational limitations, we cannot fine-tune or do grid-search for every model and dataset. Hence, across all runs, number of epochs is 750 with early-stopping if validation does not improve for 25 consecutive epochs, batch size is 64, window size $W$ is 60, learning rate is $3 \cdot 10^{-3}$ for model parameters and $10^{-2}$ for $\hat{\rho}$, both with Adam [35] optimizer. Two exceptions are AGCRN on M5-L10, which is trained with a $W = 30$ due to GPU memory limit, and all Conv-T models, which are trained with $10^{-3}$ learning rate that yields much better outcomes. Other model-specific hyperparameters, which mostly follow each original paper for that model if possible, are also fixed across all datasets. For simplicity, we set $\hat{\rho}$ as an $N$-dimensional vector when $N \geq 300$ and $\hat{\rho}$ as a scalar otherwise, assuming that small $N$ implies all series have similar characteristics. Detailed listings of hyperparameters can be found in Appendix C.

The results are shown in Table 1. In all but four cases, our method improves the performance of the model. Even in the event that our method is ineffective, the deterioration is small. Notice that the averaged relative improvement is more significant on LSTM than on AGCRN or DSANet. We conjecture that AGCRN and DSANet are better model structures for time series than LSTM, so the autocorrelated errors resulting from model misspecification are less severe. Experiments in Section 5.8 below further support this conjecture.

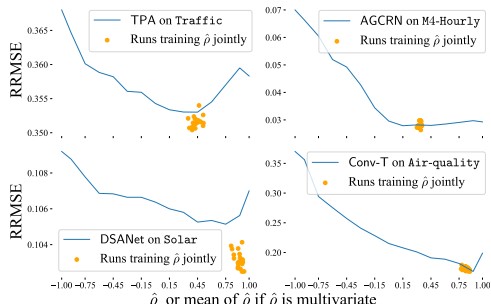
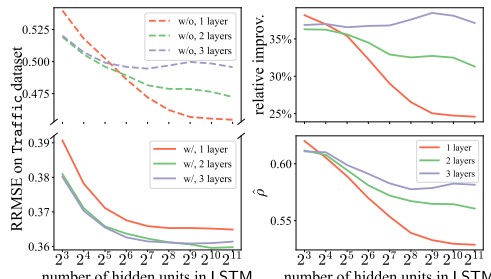

Figure 2: RRMSE of four combinations of models and datasets averaged over 20 runs when $\hat{\rho} \in [-1, 1]$ is fixed. Orange dots are the results when $\hat{\rho}$ is not fixed and is instead learned jointly with model parameter $\theta$.

Figure 3: Results of training LSTM on the Traffic dataset with different numbers of layers and hidden units averaged over 30 runs using both the w/o and w/ methods. In the upper right subfigure, $y$-axis is the relative improvement of using w/ over w/o.

## 5.5 Empirical critical values of remaining autocorrelation in errors

As introduced in Section 3.1, the Durbin-Watson statistic [16] is used in typical linear models to detect the statistical significance of autocorrelated errors. Since the specific critical values of Durbin-Watson statistic depend on the data distribution, parameter space, and the optimization process, it is impossible to obtain exact theoretical critical values for NNs on real-world datasets. Thus, as a substitute, we aim to provide empirical critical values of remaining autocorrelation in errors as a guideline to determine whether adjusting for autocorrelated errors is necessary.

We first plot all runs in Section 5.4 in the upper plot in Figure 1. For every single run, we can calculate the "remaining autocorrelation in errors" (as Equation (8) but averaged over $N$ series) and the relative improvement compared to the corresponding without-adjustment MSE in Table 1. Then, regression lines for both methods are also plotted. Note that the "remaining autocorrelation in errors" is different from the learned autocorrelation coefficient $\hat{\rho}$. Conceptually, the larger the $\hat{\rho}$, the stronger the adjustment, and thus the weaker the remaining autocorrelation. From the regression lines, we can see that adjusting for autocorrelated errors improves performance on average, and the improvement is larger when the remaining autocorrelation is weaker. This means that if the adjustment is more successful (i.e., finding better $\hat{\rho}$ so the remaining autocorrelation is close to 0), larger improvement is made.

Additionally, we compute the density estimation in the $x$-dimension as shown in the bottom plot in Figure 1. It is clear that before adjustment, there are many runs that have significant nonzero autocorrelated errors. This identifies the existence of autocorrelated errors. Moreover, after applying our method of adjustment, the density of the remaining autocorrelation is more concentrated around 0, implying our adjustment does reduce the autocorrelation in the errors. Finally, the right-tailed 1%, 5% and 10% empirical critical values of the w/o method are 0.984, 0.928, 0.857. In the future, these empirical critical values can provide guidelines to whether the adjustments are necessary. For example, observing 0.928 remaining autocorrelation without adjustment implies that there is statistical evidence that errors are autocorrelated at 95% significance. In other words, applying our method can decrease the autocorrelation and thus improve model performance 95% of the time.

## 5.6 Grid-searching hyperparameters

To further verify the advantages of our method, we select the four cases in Table 1 where our method underperforms, and grid-search hyperparameters on them. For all models, the choices are $\{10^{-3}, 3 \cdot 10^{-3}\}$ for learning rate and $\{32, 64\}$ for batch size. For AGCRN, hyperparameters are searched over $\{32, 64\}$ for hidden units and $\{2, 10\}$ for embedding dimension. For DSANet, filter length in local convolution are chosen from $\{3, 5, 7\}$ and number of layers from $\{1, 2\}$. Other hyperparameters follow those in Section 5.4.

The best RRMSEs are shown in Table 2. Out of the four combinations, our method produces better outcomes in three, which implies that the improvement is consistent across different hyperparame-

Table 3: Averaged relative improvement of three adjustment types compared to no adjustment at all. Best performance in boldface and is superscribed with † if the p-value of paired t-test between the first and second best method is lower than 5%.

| Models | TPA | | | | AGCRN | | | | DSANet | | | |
|---|---|---|---|---|---|---|---|---|---|---|---|---|
| Datasets | w/o | w/ inp. | w/ out. | w/ | w/o | w/ inp. | w/ out. | w/ | w/o | w/ inp. | w/ out. | w/ |
| PeMSD8 | .1392 | .1392 | .1392 | **.1388** | .1370 | .1375 | .1373 | **.1366**$^\dagger$ | .1409 | .1408 | .1408 | **.1398**$^\dagger$ |
| ADI-945 | .0545 | .0452 | .0425 | **.0368**$^\dagger$ | .0095 | **.0082** | .0085 | .0084 | .0709 | .0592 | .0714 | **.0581** |
| M4-Hourly | .0520 | **.0394**$^\dagger$ | .0415 | .0416 | .0282 | .0277 | **.0272**$^\dagger$ | .0284 | .0902 | .0886 | .0910 | **.0773**$^\dagger$ |
| M5-L9 | .1898 | .1915 | .1922 | **.1854**$^\dagger$ | .1959 | .1949 | .1945 | **.1925**$^\dagger$ | .1823 | **.1822** | .1823 | .1824 |
| Exchange | .0509 | .0437 | .0401 | **.0354**$^\dagger$ | .0124 | .0136 | .0130 | **.0119**$^\dagger$ | .0115 | .0115 | .0121 | **.0109**$^\dagger$ |
| Solar | .1061 | .1054 | .1061 | **.1052**$^\dagger$ | .0994 | .1002 | .0999 | **.0992** | .1064 | .1033 | .1063 | **.1032** |
| Avg. rel. impr. | | 9.23% | 10.4% | **14.4%** | | 1.00% | 1.54% | **2.86%** | | 3.52% | -1.21% | **6.86%** |

ters. Notice that AGCRN is specifically designed for traffic forecasting, which explains why it is challenging to improve on top of it when the target dataset is traffic-related, as also shown in Table 1.

## 5.7  Grid-searching $\hat{\rho}$

Similar to the Hildreth-Lu method as described in Section 3.1, we grid-search over possible values of $\hat{\rho}$ and then fix it during training in order to find the best $\hat{\rho}$. The search range is $\{-1.0, -0.9, -0.75, \ldots, 0.75, 0.9, 1.0\}$ and the values are the same across all dimensions of $\hat{\rho}$. Note that fixing $\hat{\rho} = 0$ is equivalent to the w/o method.

We run four combinations as shown in Figure 2. It is clear that fixing $\hat{\rho}$ at some nonzero value yields better results compared to the w/o method (i.e. $\hat{\rho} = 0$). This shows that adjusting for autocorrelated errors does enhance the model performance. In addition, when $\hat{\rho}$ is trained jointly with $\theta$, the learned $\hat{\rho}$ is close to the optimal $\hat{\rho}$ as found by grid-search. Plus, jointly training $\hat{\rho}$ and $\theta$ may result in even better RRMSE compared to grid-search because all trainable parameters are optimized jointly.

It is also interesting to see our method compared to the case when $\hat{\rho}$ is fixed at 1, which is equivalent to the differencing technique adopted in many time series models. However, differencing and our method have three main distinctions: (1) differencing is usually used for stationarizing time series, not for adjusting autocorrelated errors, (2) differencing fixes $\hat{\rho}$ at 1, but our method has flexible $\hat{\rho}$, and (3) our $\hat{\rho}$ is trainable with NNs. These distinctions are the reasons why our method of learning $\hat{\rho}$ is better for NNs compared to differencing, as can be seen in Figure 2.

## 5.8  Effect of model misspecification

Here, we train LSTM with different numbers of layers and hidden units on the Traffic dataset to explore the effect of model misspecification. We expect that, with the same number of layers, LSTM with more hidden units is more expressive so model misspecification is less severe, and hence adjustment for autocorrelation is less beneficial. The results are shown in Figure 3.

In the left subfigures, we see that the RRMSE decreases when the number of hidden units increases in both methods. However, the relative improvement of applying w/ over w/o decreases, as shown in the upper right subfigure. Meanwhile, in the bottom right subfigure, $\hat{\rho}$ also decreases. All these observations combined indicate that when the LSTM is more expressive, the need for adjustment reduces as $\hat{\rho}$ decreases, and the advantage of adjustment weakens as relative improvement decreases.

## 5.9  Limitations

There are two main limitations of our method. First, many neural forecasting models employ complex forecasting distribution, including probabilistic forecasting [23, 36, 51, 52, 53, 54]. Our method is not applicable when the errors do not follow Equation (4). Though, note that our method can be applied with quantile regression [23, 61] by viewing the series corresponding to the target quantile as a new time series.

Second, the aforementioned phenomenon in Section 5.8 can be considered as another limitation of our method. In the extreme case, if the model is well-designed for the dataset, errors should be completely uncorrelated assuming model misspecification is the only source of autocorrelation — and only then

would our method be non-beneficial. However, in the real-world, we rarely know the DGP, so it is nearly impossible to have zero autocorrelation. Even worse, we cannot calculate the significance of the autocorrelation when using NNs. Compared to the procedure for linear models where adjustments are made only when the Durbin-Watson statistic [16] shows significant autocorrelation, we can employ the empirical critical values in Section 5.5 and adjust autocorrelated errors if necessary.

### 5.10    Ablation study

In our adjustment for autocorrelated errors for time series forecasting, we adjust for both the input and output in Equation (11). Here, we study the effect of adjusting for only input or output — that is, fixing $\hat{\rho}$ to 0 for the input or output part. The results are shown in Table 3, which is a subset of Table 1 due to computational limitations. From the table, we see that adding either input or output adjustment improves performance on average. Nevertheless, adding both gives the best overall improvement.

## 6    Experiments on Other Time Series Tasks

We also test our method on time series regression, time series classification, and unsupervised anomaly detection. Due to page limitations, we detail these experiments in the appendix. Appendix D describes the setups, models, datasets, metrics, and results on time series regression. Similarly, Appendix E and Appendix F are on time series classification and unsupervised anomaly detection, respectively. Overall, we again observe that adjusting for autocorrelated errors maintains or improves performances in most cases.

## 7    Conclusion and Broader Impacts

In this paper, we propose to adjust for autocorrelated errors by learning the autocorrelation coefficient jointly with the neural network parameters. Experimental results on time series forecasting demonstrate that applying our method in existing state-of-the-art models further enhances their performance across a wide variety of datasets. Additionally, we provide empirical critical values of remaining autocorrelation in errors to act as a guideline to determine whether adjusting for autocorrelated errors is necessary. Supplementary experiments are conducted to showcase the need for adjustment and to support the advantages of our method. Finally, we also demonstrate that our method is applicable to many other time series tasks as well.

The broader impacts lies in two aspects. First, our method can be applied on any neural networks for time series. This makes our method a useful part of the procedure of training neural networks on time series; analogous to how the Durbin-Watson statistic is a part of the procedure in ordinary least squares regression. Second, our method improves neural networks on many time series tasks. As our method is a generic algorithm for improving neural networks, it is unlikely to have a direct negative societal impact in the near future.

For future research directions, we can explore more complex, higher-order autocorrelated errors with quantile regression and probabilistic forecasting.

### Acknowledgements and Disclosure of Funding

We thank John Yamartino and Ramana Veerasingam from Lam Research and Hilaf Hasson from Amazon for their helpful discussions and insights. This work was supported in part by an unrestricted gift to MIT from Lam Research.

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
