# Appendices

## A Derivation of covariance for first-order autocorrelated errors

From $e_t = \rho e_{t-1} + \epsilon_t$, we have

$$e_t = \rho(\rho e_{t-2} + \epsilon_{t-1}) + \epsilon_t \tag{14}$$

$$= \rho^2 e_{t-2} + \rho \epsilon_{t-1} + \epsilon_t \tag{15}$$

$$= \cdots \tag{16}$$

$$= \rho^h e_{t-h} + \sum_{i=0}^{h-1} \rho^i \epsilon_{t-i}. \tag{17}$$

For any time step $t$, we can assume that there are infinite previous time steps before it (although the data is finite). Thus,

$$e_t = \lim_{h \to \infty} (\rho^h e_{t-h} + \sum_{i=0}^{h-1} \rho^i \epsilon_{t-i}). \tag{18}$$

Notice that if $\rho \geq 1$, $e_t$ explodes to infinity because $\lim_{h \to \infty} \rho^h = \infty$, so we assume $|\rho| < 1$. Then,

$$e_t = \lim_{h \to \infty} \sum_{i=0}^{h-1} \rho^i \epsilon_{t-i}. \tag{19}$$

and the variance of $e_t$ is:

$$\text{Var}(e_t) = \lim_{h \to \infty} \sum_{i=0}^{h-1} \text{Var}(\rho^i \epsilon_{t-i}) \tag{20}$$

$$= \lim_{h \to \infty} \sum_{i=0}^{h-1} \rho^{2i} \sigma^2 \tag{21}$$

$$= \frac{1}{1-\rho^2} \sigma^2. \tag{22}$$

Finally, combining Equation (17) and Equation (22), the covariance of $x_t$ and $x_{t+h}$ is:

$$\text{Cov}(e_t, e_{t-h}) = \text{Cov}(\rho^h e_{t-h} + \sum_{i=0}^{h-1} \rho^i \epsilon_{t-i}, e_{t-h}) \tag{23}$$

$$= \rho^h \text{Cov}(e_{t-h}, e_{t-h}) \tag{24}$$

$$= \frac{\rho^h}{1-\rho^2} \sigma^2, \tag{25}$$

where the final line follows from $\text{Cov}(e_{t-h}, \epsilon_{t-i}) = 0, \forall i \leq h - 1$ because $\epsilon_{t-i}$ is a Gaussian noise that happens after time step $t - h$.

## B Detailed descriptions and statistics of real-world datasets

- `PeMSD4`: Traffic data in San Francisco Bay Area from January 2018 to February 2018. Only the total traffic flow series are used.

- `PeMSD8`: Similar to `PeMSD4`, but recorded in San Bernardino from July 2016 to August 2016.

- `Traffic` [1]: Data from the California Department of Transportation describing road occupancy rates, a number between 0 and 1, of the San Francisco Bay area freeways from 2015 to 2016.

---

[1]http://pems.dot.ca

- `ADI-920`, `ADI-945`: Sensor values recorded from the plasma etcher machine in Analog Device Inc. The raw data is in three-dimension because it is split up by wafer cycles, so we concatenate all wafer cycles together and batch it without using data points from different wafer cycles. `ADI-920` and `ADI-945` represent two different recipes.

- `M4-Hourly`, `M4-Daily`, `M4-Weekly`, `M4-Monthly`, `M4-Quarterly`, `M4-Yearly`: Six datasets from the M4 competition with different sampling rate. Each dataset contains miscellaneous series, categorized into six domains (micro, industry, macro, finance, demographic, other). Originally, each series has different length and start time, but we crop out part of each dataset so every series has the same start time and length without missing value.

- `M5-L9`, `M5-L10`: Raw data are Walmart unit sales of 3,049 products sold in ten stores in three States (CA, TX, WI). The products can be categorized into 3 product categories (Hobbies, Foods, and Household) and 7 product departments. `M5-L9` is the level-9 aggregation and `M5-L10` is the level-10 aggregation [2].

- `Air-quality` [3]: Recorded by gas multisensor devices deployed on the field in an Italian city.

- `Electricity` [4]: Electricity consumption in kWh from 2012 to 2014.

- `Exchange`: exchange rate of eight countries (Australia, British, Canada, Switzerland, China, Japan, New Zealand, and Singapore) from 1990 to 2016.

- `Solar` [5]: Solar power production records in 2006 from photovoltaic power plants in Alabama State.

| Datasets | length of dataset $T$ | number of series $N$ | sampling rate |
|---|---|---|---|
| PeMSD4 | 16,992 | 307 | 5 minute |
| PeMSD8 | 17,856 | 170 | 5 minute |
| Traffic | 17,544 | 862 | 1 hour |
| ADI-920 | 288,098 | 30 | 2.5 seconds |
| ADI-945 | 103,073 | 30 | 2.5 seconds |
| M4-Hourly | 744 | 121 | 1 hour |
| M4-Daily | 4,208 | 1,493 | 1 day |
| M4-Weekly | 2,191 | 43 | 1 week |
| M4-Monthly | 816 | 203 | 1 month |
| M4-Quarterly | 674 | 27 | 1 quarter |
| M4-Yearly | 618 | 9 | 1 year |
| M5-L9 | 1,941 | 70 | 1 day |
| M5-L10 | 1,941 | 3049 | 1 day |
| Air-quality | 9,357 | 13 | 1 hour |
| Electricity | 26,304 | 321 | 1 hour |
| Exchange | 7,588 | 8 | 1 day |
| Solar | 52,560 | 137 | 10 minutes |

## C  Hyperparameters of all NNs for time series forecasting

LSTM: number of layers = 2, hidden size = 64.

TPA: number of layers = 1, hidden size = 64, linear autoregressive size = 24.

AGCRN: number of layers = 1, hidden size = 64, embedding dimension = 10.

TCN: number of layers = 9, hidden size = 64.

Conv-T: number of layers = 3, number of attention head = 8, hidden size = 256, filter kernel size = 6.

DSANet: number of layers = 1, local filter size = 3, number of channels = 32, dropout = 0.1.

---

[2]Please see the competition guidelines (https://mofc.unic.ac.cy/m5-competition) for definition of aggregations.

[3]https://archive.ics.uci.edu/ml/datasets/Air+quality

[4]https://archive.ics.uci.edu/ml/datasets/ElectricityLoadDiagrams2011

[5]http://www.nrel.gov/grid/solar-power-data

# D Time series regression

## D.1 Task description

In time series regression, the target variable $y_t \in \mathbb{R}$ at time step $t$ only depends on the input variables $\mathbf{X}_t \in \mathbb{R}^N$ at the same time step $t$. Thus, a time series regression dataset consists of $T$ input-target pairs: $\{(\mathbf{X}_1, y_1), \ldots, (\mathbf{X}_T, y_T)\}$. The goal is to find the optimal $\theta$ to minimize the MSE:

$$\text{MSE} = \sum_{t=1}^{T} e_t^2 = \sum_{m=1}^{T} (y_t - \hat{y}_t)^2, \tag{26}$$

where $\hat{y}_t = f(\mathbf{X}_t; \theta)$ is the prediction of the model.

## D.2 Datasets

We synthesize our data using Monte Carlo simulations. The data-generating function is

$$
\begin{aligned}
y_t &= \tanh(\frac{\mathbf{X}_t \theta + 1}{\sqrt{N}}) + e_t, \quad \mathbf{X}_t \in \mathbb{R}^N, \theta = \mathbf{1} \in \mathbb{R}^N, \\
\mathbf{X}_t &\sim \mathcal{N}(\mathbf{0}, \sigma_x^2 \mathbf{I}), \quad \sigma_x = 0.2, \\
e_t &= \rho e_{t-1} + \epsilon_t, \quad \epsilon_t \sim \mathcal{N}(0, \sigma^2).
\end{aligned}
\tag{27}
$$

To compare different methods across a diverse set of datasets, we synthesize 30 random datasets for each combination of all the following values:

- $T \in \{25, 50, 100, 200, 400\}$,
- $N \in \{2, 3, 6, 12, 24\}$,
- $\rho \in \{-0.9, -0.75, \ldots, 0.75, 0.9\}$, and
- $\sigma \in \{0.0025, 0.005, 0.01, 0.02, 0.04\}$.

Thus, there are a total of $48,750$ runs for each method.

For each synthesized training set with $T$ samples, we synthesize $100T$ samples as the testing set. $20\%$ of the training set are split into the validation set, and the model with the best validation MSE with 750 epochs is chosen. When $N$ is small, to avoid good or bad luck, the validation MSE of the first 5 epochs are ignored.

## D.3 Models

The NN we use has six fully-connected layers with ReLU activation function and three residual connections. The learning rate is $5 \cdot 10^{-3}$ for $\theta$ and $10^{-2}$ for $\hat{\rho}$, both with Adam optimizer.

## D.4 Results

There are three methods to be compared. First is the without adjustment method, denoted as w/o. Next is the modified Prais-Winsten method, which is described in the beginning in Section 4, denoted as mPW. Finally is our method of adjustment, denoted as w/.

Pairwise comparison of w/, mPW, and w/o are shown in the subfigures in Figure 4. From the figures, we see that on average, w/ is better than mPW, which is better than w/o. Moreover, looking at the left subfigure, we see that the outperformance is greater when the true $\rho$ value has larger magnitude. Particularly, when $\text{abs}(\rho) > 0.15$, w/ beats mPW and w/o with statistical significance, and when $\text{abs}(\rho) \le 0.15$, mPW is the best method on average, but all three methods have similar performances. Finally, observing the right subfigure in Figure 4, we see that adjusting for autocorrelated errors, using either w/ or mPW has greater improvement when the noise has a larger scale.

In Figure 5, the values of $\text{abs}(\rho - \hat{\rho})$ averaged over all runs are shown. Similar to the outcomes in Figure 4, we see that w/ has better estimates of $\rho$ when $\text{abs}(\rho) > 0.15$, and mPW has the upper hand otherwise. The reason, we believe, lies in step 2 in the mPW method. When the scale of $\rho$ is small, there is little benefit in adjusting for autocorrelation, so the first iteration of step 2 in mPW will

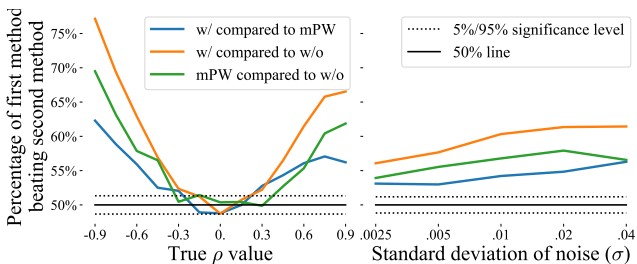

Figure 4: Pairwise comparison of three methods of training NN on the synthesized data with different true $\rho$ value (left) and different standard deviation of noise (right).

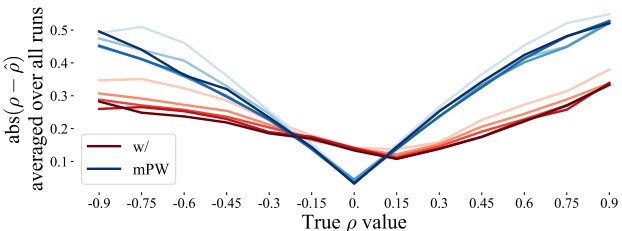

Figure 5: abs$(\rho - \hat{\rho})$ of neural network averaged over all runs for "w/" and "mPW" on the synthesized data. Lines with higher opacity represent results that are run on data with larger variance of noise.

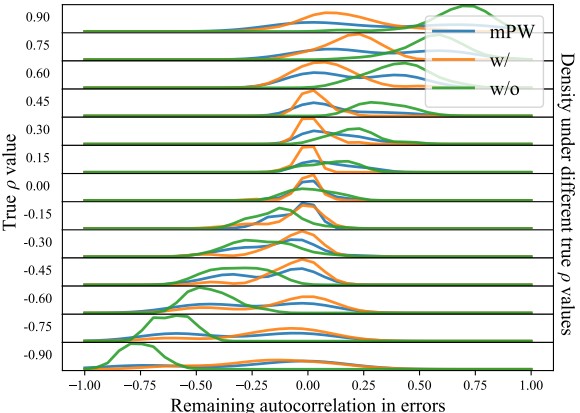

Figure 6: The distribution of Durbin-Watson statistic under different data-generation processes. Each row correspond to different true $\rho$ value. The $x$-axis is the Durbin-Watson statistic from $0$ to $4$ and the $y$-axis is the density of the distribution calculated over $3,750$ runs.

essentially train the model to near convergence just like the w/o method and will result in small-scale $\hat{\rho}$. In contrast, training $(\theta, \hat{\rho})$ together from the onset, as in w/, might result in a larger-scale $\hat{\rho}$ because the NN is not yet converged in the beginning.

### D.5    Decrease in autocorrelation

After adjusting for autocorrelated errors, we expect the remaining autocorrelation in errors to stay close to $0$ no matter what the true $\rho$ value is. Both mPW and w/ do exhibit such phenomenon as shown in Figure 6. In the figure, the distribution of the remaining autocorrelation shifts from around $-1$ to around $1$ when no adjustment is made. Applying either mPW or w/ stabilize the distribution around $0$, especially using w/ because its distribution is more concentrated around $0$

Table 4: Statistics of the time series classification datasets.

| Datasets | number of classes | number of series | max training length | train-test split |
|---|---|---|---|---|
| Spoken Arabic Digits | 10 | 13 | 93 | 75-25 split |
| Australian Sign Language | 95 | 22 | 96 | 44-56 split |
| ECG | 2 | 2 | 147 | 50-50 split |
| Pen-based Digits | 10 | 2 | 8 | 2-98 split |

Table 5: Accuracy and average relative improvement of all combinations of models and datasets averaged over twenty runs. w/o implies without adjustment for autocorrelated errors, whereas w/ implies with adjustment. Best performance is in boldface and is superscribed with † if the p-value of paired t-test is lower than $5\%$. Average relative improvement is the percentage of improvement of w/ over w/o averaged over all datasets for each model.

| Models | MLSTM-FCN | | FCN-SNLST | |
|---|---|---|---|---|
| Datasets | w/o | w/ | w/o | w/ |
| Spoken Arabic Digits | **0.9929** | 0.9922 | **0.9935** | 0.9925 |
| Australian Sign Language | 0.9388 | **0.9446** | **0.9924** | 0.9918 |
| ECG | 0.8100 | **0.8300** | 0.8575 | **0.8670**† |
| Pen-based Digits | 0.9609 | **0.9631**† | 0.9530 | **0.9542** |
| Avg. rel. improv. | | 0.81% | | 0.27% |

# E  Time series classification

## E.1  Task description

In time series classification, the dataset consists of $M$ time series data $\{\mathbf{X}^{(1)}, \ldots, \mathbf{X}^{(m)}\}$ and we want to classify each time series data into $K$ classes. In other words, a target $y^{(m)} \in [1, K]$ is the class label for the time series data $\mathbf{X}^{(m)}$. The goal is to find the optimal $\theta$ to maximize accuracy (ACC):

$$\text{ACC} = \frac{\sum_{m=1}^{M} \mathbb{1}\{y^{(m)} = \hat{y}^{(m)}\}}{M} . \tag{28}$$

## E.2  Datasets

We include four datasets from the Baydogan archive [45].

- Spoken Arabic Digits contains time series of mel-frequency cepstrum coefficients (i.e., frequency-domain speech signals) corresponding to spoken Arabic digits from 44 male and 44 female native Arabic speakers;

- Australian Sign Language consists of high-quality position trackings on 27 native signers' hands for 95 signs;

- ECG contains electrocardiogram signals to classify between a normal heartbeat and a Myocardial Infarction;

- Pen-based Digits consists of $x$ and $y$-coordinates of hand-written digits on a pressure sensitive tablet by 44 writers.

The statistics of the datasets can be found in Table 4

## E.3  Models

Two state-of-the-art models are chosen:

- Multivariate Long Short-Term Memory Fully Convolutional Network (MLSTM-FCN) [34],

- Fully Convolutional Network with Stacked Neural Low-rank Sequence-to-Tensor transformations (FCN-SNLST) [59].

Table 6: Statistics of the unsupervised anomaly detection datasets.

| Datasets | number of series | max training length | sampling frequency | input signals |
|---|---|---|---|---|
| ADI Etcher | 30 | 12,644 | 2 | Mixed |
| CWRU Bearing | 2 | 485,643 | 1200 | Vibration |
| Chiron Mill | 3 | 98,324 | 1000 | Vibration |
| Harting Mill | 3 | 55,137 | 200 | Vibration |

| Hyperparameters | ADI Etcher | CWRU Bearing | Chiron Mill | Harting Mill |
|---|---|---|---|---|
| batch size | 128 | 128 | 16 | 128 |
| Learning rate | 2e-4 | 1e-4 | 1e-4 | 1e-4 |
| Epochs | 50 | 50 | 50 | 50 |
| Smooth size | 1 | 1 | 128 | 1 |
| Down-sample | 1 | 1 | 2 | 1 |
| STFT length | 64 | 8 | 1024 | 32 |
| Window size | 8 | 4 | 23 | 64 |
| Dropout rate | 0.5 | 0.5 | 0 | 0 |
| Number of GMMs | 8 | 2 | 2 | 8 |
| Model layers | 3 | 1 | 4 | 6 |
| Model size | 256 | 2 | 512 | 128 |
| Latent dimension | 64 | 32 | 48 | 32 |
| $\lambda_e$ | 0.03 | 0.03 | 0.5 | 0.03 |
| $\hat{\rho}$ dimension | 1 | 1 | 1 | 1 |
| Threshold type | max | mean | max | mean |

Table 7: Hyperparameters of the TC-DAGMM model on each dataset.

### E.4 Results

We closely follow the hyperparameters in the original papers [34, 59]. MLSTM-FCN is trained for 1000 epochs with batch size 128. Seq2Tens is trained until loss does not decrease for 300 epochs with batch size 4. We adjust for autocorrelated errors in the input series and set $\hat{\rho}$ as a scalar in all cases. The results are shown in Table 5. In addition to accuracy (ACC), to aggregate results over multiple datasets, the average relative improvement

$$\frac{1}{D}\sum_{d=1}^{D}\frac{(\overline{\mathrm{ACC}}_{\mathrm{w/,\,d}} - \overline{\mathrm{ACC}}_{\mathrm{w/o,\,d}})}{\overline{\mathrm{ACC}}_{\mathrm{w/o,\,d}}} \cdot 100\% \tag{29}$$

is also reported, where $D$ is the number of datasets, w/o denotes training without adjustment, w/ denotes with adjustment, and $\overline{\mathrm{ACC}}$ is the averaged ACC over multiple runs.

From the table, we can observe that adjusting for autocorrelated errors is beneficial in most cases, especially when there is room for improvement. For instance, since the accuracies of both models on Spoken Arabic Digits and FCN-SNLST on Australian Sign Language are already very close to 1, adjustments in these cases are not helpful. But for other cases, applying our method does help. From the table, we can conclude that if the accuracy without adjustment reaches 0.99, then adjusting for autocorrelated erorrs is not necessary.

Although our method is also effective on time series classification, the improvement is much smaller when compared to the results in time series forecasting. The average relative improvements are much smaller and the number of statistical significant cases are also lower. This is expected because the target in classification is a discrete class label and the label is for a time series instead of just one time step.

# F Unsupervised anomaly detection

## F.1 Task description

Unsupervised anomaly detection is similar to time series classification because a time series instance is either classified as a "normal" one or an anomaly. However, the difference is that the model is trained in an unsupervised manner — it only has access to the normal data but not the anomalous data. Thus, in unsupervised anomaly detection, a *training* dataset consists of only the $M$ normal inputs $\{\mathbf{X}^{(1)}, \ldots, \mathbf{X}^{(M)}\}$. In testing, input-target pairs that includes both normal and anomalous data are provided to measure the performance of the model.

Similar to time series classification, given $f$ and $\theta$, accuracy (ACC) is one of the metrics we use. Another metric is the area under curve (AUC) of the receiver operating characteristic (ROC) curve, which is a more straightforward overall metric for unsupervised training.

## F.2 Datasets

For unsupervised anomaly detection, we focus on four manufacturing datasets following [44]:

- `ADI Etcher` [12, 28, 41]: internal sensor readings from a Lam Research plasma etcher in a production line at ADI. A fault is defined when the etched wafer fails the subsequent electronic test.
- `CWRU Bearing` [40]: vibration data from a spinning fan supported by circular ball bearings collected by Case Western Reserve University (CWRU). Faults occurr when defects are introduced to the inner raceway, outer raceway, or balls in the bearing using electro-discharge machining.
- `Chiron Mill`: axis accelerometer data from a Chiron milling machine. The normal data uses new tools whereas the anomalies use worn tools.
- `Harting Mill`: three axis vibration data from a milling machine collected by Harting. Faults are introduced by switching off the cooling system in the milling machine.

The statistics of the datasets are shown in Table 6.

## F.3 Models

Temporal Convolutional Deep Autoencoding Gaussian Mixture Model (TC-DAGMM) is used. This model is based on the Deep Autoencoding Gaussian Mixture Model (DAGMM) [65] framework, but several changes are made so that the model can be successfully trained on the manufacturing data. The changes that influence the training process include

- We adopt Temporal Convolutional Networks (TCN) as the encoder in the DAGMM framework so that the model can look at multiple time steps per example.
- We adopt transposed convolution TCN as the decoder in the DAGMM framework.
- We do not let the Estimation Network in the DAGMM framework estimate the covariance matrix for the Gaussian Mixture Model (GMM). Instead, we fix the covariance matrix as an identity matrix. Consequently, the error term involving the covariance matrix is eliminated.
- Shortening transformations (smoothing, down-sampling, and short-time Fourier transform (STFT)) are applied on the raw data to deal with lengthy time series.
- Sliding window with size $W$ slices the shortened time series and the model only looks at one slice at a time.

If we use $T^{(m)}$ to denote the length of the time series data $\mathbf{X}^{(m)}$ after shortening, then there are $T^{(m)} - W + 1$ anomaly scores $s_t^{(m)}$, each for one slice of $\mathbf{X}^{(m)}$. That is,

$$s_t^{(m)} = f(\mathbf{X}_{t:t+W-1}^{(m)}; \theta), \ \forall t \in [1, T^{(m)} - W + 1] . \tag{30}$$

We define the anomaly score $s^{(m)}$ for the whole $\mathbf{X}^{(m)}$ by either the mean or the max of the $T^{(m)} - W + 1$ anomaly scores:

$$s^{(m)} = \mathrm{mean}(\{s_t^{(m)}\}_{t=1}^{T^{(m)}-W+1}) \quad \text{or} \quad s^{(m)} = \mathrm{max}(\{s_t^{(m)}\}_{t=1}^{T^{(m)}-W+1}) . \tag{31}$$

| Accuracy | | |
|---|---|---|
| Datasets | w/o | w/ |
| ADI Etcher | 0.9227 | **0.9252**$^{\dagger}$ |
| CWRU Bearing | **0.9857** | 0.9786 |
| Chiron Mill | 0.7333 | **0.7625**$^{\dagger}$ |
| Harting Mill | 0.9379 | **0.9481**$^{\dagger}$ |
| Avg. rel. improv. | | 1.15% |

| Area under curve | | |
|---|---|---|
| Datasets | w/o | w/ |
| ADI Etcher | **0.9585**$^{\dagger}$ | 0.9525 |
| CWRU Bearing | **1.000** | **1.000** |
| Chiron Mill | 0.7222 | **0.7593**$^{\dagger}$ |
| Harting Mill | **0.9977** | **0.9977** |
| Avg. rel. improv. | | 1.14% |

Table 8: Accuracy (upper), area under curve (lower), and average relative improvement of TC-DAGMM on all datasets averaged over twenty runs. w/o implies without adjustment for autocorrelated errors, whereas w/ implies with adjustment. Best performance is in boldface and is superscribed with † if the p-value of paired t-test is lower than $5\%$. Average relative improvement is the percentage of improvement of w/ over w/o averaged over all datasets for each model.

Finally, during testing, the anomaly threshold is set as two standard deviations from the mean of $M$ anomaly scores in the training set:

$$\text{anomaly threshold } = \text{mean}(\{s^{(m)}\}_{m=1}^{M}) + 2 \text{ std}(\{s^{(m)}\}_{m=1}^{M}) . \tag{32}$$

That is, any time series data $\mathbf{X}^{(\bullet)}$ is classified as an anomaly if the anomaly score $s^{(\bullet)}$ is higher than the anomaly threshold during test time.

### F.4 Hyperparameters

Manufacturing datasets exhibit different characteristics from one dataset to another. Empirically, it is difficult to train a NN on different manufacturing datasets with the same hyperparameters. Thus, we tune the hyperparameters for each dataset to achieve the best possible results, but the hyperparameters are identical within the same dataset with or without adjustment. The tuned hyperparameters are listed in Table 7, where $\lambda_e$ is the weight of energy loss in the DAGMM framework.

### F.5 Results

The results are shown in Table 8. In addition to accuracy (ACC) and area under curve (AUC), we also report the average relative improvement on both metrics, defined similar to equation (29).

From the table, we can observe that adjusting for autocorrelated errors is beneficial or at least harmless in most cases. Analogous to the cases in time series classification, adjustment is more helpful when there is room of improvement in the base model. The relative improvement is slightly better than that in time series classification, probably because the DAGMM framework contains an autoencoding structure.