# OpenReview forum: "Adjusting for Autocorrelated Errors in Neural Networks for Time Series"
_NeurIPS.cc/2021/Conference — NeurIPS 2021 Poster_

### Official Review · Reviewer_PpHM · 2021-07-06

**Rating:** 6
**Confidence:** 2

**Summary:**

The authors consider the problem of controlling for autocorrelated errors in neural networks. This is accomplished through the introduction of a new training scheme that jointly learns the first-order autocorrelation coefficient alongside model parameters.

This is backed up by a lot of experiments that, particularly for time series forecasting, demonstrate significantly improved performance almost across the board by implementing the new training procedure.

**Limitations And Societal Impact:**

Yes.

**Main Review:**

## Strengths
- A nice, simple approach to solve a (surprisingly) little-studied problem of dealing with autocorrelated errors in time series with NN models achieving excellent performance improvements.
- The quality of experiments and performances is extremely good, in particular for time series regression problems. The joint training of $\rho$ with model parameters seems to offer clear benefits and the additional studies are good.
- Paper is generally clear and well written.

## Weaknesses
**Method explanation** I found the discussion on the method section to be rather confusing. This could just be me, but step 2 made me scratch my head as to what was going on. I assume you are minimising the residuals after adjusting for the autocorrelated errors, that is:

minimise over $\theta, \sum \quad X_t - \rho X_{t-1} - (f_{\theta, t} - \rho f_{\theta, t-1})$

I may have this wrong, but either way, it is not clear in the method section what is going on. I think it could do with a rewrite and would add a lot to the paper.

**Inclusion of $\bar{X}$** It is stated in lines **190-191** that $X_{t - W - 1}$ is removed since this would include an extra timestep not used in the comparison model. However, this is replaced by the training mean, $\bar{X}$, why is this done rather than removal of the step? Surely the point of removing $X_{t - W - 1}$ was so we can have a fair comparison to the standard model using $W$ time steps, but now more information has been added through $\bar{X}$ that is not seen in the 'w/o' model slightly muddying the comparison.


**Time Spent Reviewing:**

3

---

> ### Author Response · Authors · 2021-08-09
> **Response to Reviewer PpHM**
>
> We thank the reviewer for the thoughtful review, comments, and constructive feedback. We provide answers to the comments below.
>
> Q1:  I found the discussion on the method section to be rather confusing.
>
> A1:
> We agree that the method section is challenging and can be confusing, but with good reasons. The method we derive is a reformulation of the original optimization problem. The method is developed both mathematically and empirically, with reasonable rationale as described in the paper. There can possibly be many ways of reformulation and we cannot prove that our method is the best. However, although not mentioned in the paper, we have tried several other possible approaches, but the method in the paper is the only one we have found that achieves outstanding results while also being relatively simple and with reasonable rationale. Without the method, as mentioned in line 170, in our explorations the NNs could not be successfully trained.
>
> Q2: This could just be me, but step 2 made me scratch my head as to what was going on.
>
> A2:
> Yes, you are right. The naive procedure is minimizing the errors after adjusting for autocorrelated errors. We can clarify the paper on that point.
>
> Q3: Inclusion of $\bar{\mathbf{X}}$
>
> A3:
> Yes, you are right that $\bar{\mathbf{X}}$ slightly muddies the comparison. However, this is a valid operation because we still only use information from the training dataset. While using information further processed from the training dataset is always a choice, the key to success is how to use it. Just like how batch normalization also uses training data more fully to improve training, our method also uses $\bar{\mathbf{X}}$ to improve training. Finally, note that the $\bar{\mathbf{X}}$ is averaged over the whole training dataset, and this provides rather limited addition useful information; in other words, it is more like a constant than an informative input to the model.

---

### Official Review · Reviewer_x1qr · 2021-07-16

**Rating:** 7
**Confidence:** 5

**Summary:**

In this study, authors propose a loss function that learns the autocorrelation coefficient of time series data jointly with the model parameters to adjust the autocorrelated errors that may present in neural network-based time series forecasting algorithms.

**Limitations And Societal Impact:**

Limitations of the proposed method and the Broader Societal Impacts have discussed

**Main Review:**

Strengths

•	The motivation of the study is clear with a clear introduction to the three sources of error in machine learning.

•	Authors have used six deep learning-based architectures (covering all major neural network architectures) for time series forecasting to apply the proposed autocorrelation adjustment and evaluated these methods using multiple publicly available benchmark datasets. Authors have also shown the statistical significance of the results.

•	The correct use of relative and a scale-invariant error measure to evaluate datasets where sets of many time series are available

•	An additional ablation study conducted to further analyse the effect of adjusting the input and output for autocorrelated errors.

•	Additional experiments have conducted (Section 5.5) to justify that there exist residuals in the baseline model, and the proposed adjustment procedure reduce the remaining autocorrelation error.

Weaknesses

•	If possible, in addition to the neural network-based benchmarks, it’s always a good idea to have some statistical time series forecasting methods to include in your experiments (ARIMA, Prophet, or TBATS)


**Time Spent Reviewing:**

2 hours

---

> ### Author Response · Authors · 2021-08-09
> **Response to Reviewer x1qr**
>
> Thanks to the reviewer for the helpful review, comments, and constructive feedback. We provide answers to the comments below.
>
> Q1: In addition to the neural network-based benchmarks, it’s always a good idea to have some statistical time series forecasting methods to include in your experiments.
>
> A1:
> Our work focuses on adjusting autocorrelated errors in NNs because (1) there exist adjusting methods for some statistical methods as mentioned in Section 3.1, (2) our method can be used only with gradient descent, thus is not applicable to most statistical methods, and most importantly, (3) this work focuses on adjusting for NNs, so the experiments only contain pair comparison on NNs. An interesting area for future work might be a broader paper comparing NN and statistical time series forecasting methods, both with their respective state-of-the-art autocorrelation correction methods (i.e., our proposed new state-of-the-art method specifically for NNs, against statistical methods with their correction methods noted in Section 3.1).

---

### Official Review · Reviewer_3yo9 · 2021-07-16

**Rating:** 7
**Confidence:** 2

**Summary:**

This paper proposes a strategy for accounting for autocorrelated errors in time series that can be used with a variety of existing neural nets. Essentially we learn a neural net’s parameters jointly with an additional autocorrelation coefficient parameter that is based on a first-order autocorrelation error approximation.

**Limitations And Societal Impact:**

See the second bullet point in "weaknesses" from my main review.

**Main Review:**

This paper cleverly combines insights from how autocorrelation error is dealt with in econometrics with neural nets for time series analysis. Overall, I found the paper to be easy to follow, and the proposed method to be a major ML advance (since it’s able to improve a vast number of existing neural net models, making them handle autocorrelated errors across time) albeit with pieces that are not fully justified/well-understood. I’ve listed strengths and weaknesses below.

Strengths:
- This paper tackles an extremely common problem in time series analysis tasks (that errors across time are very commonly correlated rather than independent)
- This paper combines insights from econometrics with recent neural net advances in a clever way
- The experimental results are quite strong and suggest that accounting for autocorrelation error in time series analysis typically improves prediction accuracy

Weaknesses:
- Some of the algorithmic edits (lines 177-187) to the naive procedure described in Section 4 come off as somewhat ad hoc; while they have some justification, precise details of what is happening theoretically/under the hood is a bit unclear to me (could the problem setup have been formulated in a way where these edits just naturally appear as opposed to being introduced as modifications?). Also, perhaps an ablation study of sorts could be used to explain how much each of the edits helps improve results.
- I don’t think the last statement on page 9 is true in that if a neural net is used for some application that has negative societal impact (and where higher accuracy turns out to be correlated to more negative societal impact), then the proposed method could potentially exacerbate the negative impact.

**Time Spent Reviewing:**

3

---

> ### Author Response · Authors · 2021-08-09
> **Response to Reviewer 3yo9**
>
> We thank the reviewer for the helpful review, comments, and constructive feedback. We provide answers to the comments below.
>
> Q1: Some of the algorithmic edits (lines 177-187) to the naive procedure described in Section 4 come off as somewhat ad hoc.
>
> A1:
> The reviewer is correct that these are not a provably “best” way to improve the optimization, and have a somewhat exploratory flavor. The solutions 2 and 3 mentioned on lines 177-187 are a reformulation of the original optimization problem. They are developed to improve the optimization of the naive procedure, just like different methods are developed to better adjust for autocorrelated errors in linear model as introduced in Section 3.1. As mentioned in line 170, we found that the NNs could not be successfully trained without these solutions. Although not mentioned in the paper, we have tried several other possible ways to improve the optimization, but the approach in the paper is the only one that achieves outstanding results while being relatively simple and with reasonable rationale. We hope our work might inspire future efforts to further discover or systematize an optimal approach.
>
> Q2:
> Perhaps an ablation study of sorts could be used to explain how much each of the edits helps improve results.
>
> A2:
> That is a good suggestion, and we considered an ablation study which can be useful to filter or identify the degree of improvements from multiple components. However, an ablation study is not effective in our case. Without solution 2 or 3, the performance greatly falls behind even the without-adjustment method: the model does not learn well, let alone adjust for autocorrelated errors.
>
> Q3: I don’t think the last statement on page 9 is true in that if a neural net is used for some application that has negative societal impact.
>
> A3: Certainly it is possible that our work can have a negative societal impact. However, we use the word “direct” in the last statement to emphasize that our work cannot be used in isolation and can only have an indirect negative societal impact if used in combination with some negative-impact NN.

---

### Official Review · Reviewer_vjG1 · 2021-07-18

**Rating:** 5
**Confidence:** 4

**Summary:**

The paper argues for handling auto-correlated errors in time series forecasting, in particular for neural network and proposes methods to adjust for such auto-correlated errors via adjusting standard MSE to incorporate auto-correlated errors. In extensive empirical studies, the authors show that their approach improves the accuracy of methods (and as a by-product show the general applicability).

**Limitations And Societal Impact:**

Yes

**Main Review:**

While I agree that taking auto-correlated errors into account is an important topic that hasn't received the attention it deserves in the ML literature and the authors build a convincing case for their technique, I'm concerned with a number of aspects that I'll try to outline in the following:

l. 85: "MSE is equivalent to MLE." This is only true if we assume Gaussianity and is at best sloppy writing if taken isolatedly. However, It points to a larger issue which is an apparent unfamiliarity of the authors with the state of the art in probabilistic neural forecasting. For example, much of the modern neural forecasting literature is concerned with non-Gaussian errors since the first published neural forecasting models such as https://www.sciencedirect.com/science/article/pii/S0169207019301888, there are quantile regression approaches
https://arxiv.org/pdf/1711.11053.pdf http://proceedings.mlr.press/v89/gasthaus19a/gasthaus19a.pdf How does the method presented here extend to such non-MSE-based approaches? This needs a discussion since so much of the state of the art is concerned with lifting Gaussianity assumptions.

l. 88: there is quite some recent work on multivariate probabilistic forecasting models with neural networks:
https://arxiv.org/abs/2002.06103
https://papers.nips.cc/paper/2019/hash/0b105cf1504c4e241fcc6d519ea962fb-Abstract.html
https://arxiv.org/abs/2101.12072
https://proceedings.neurips.cc/paper/2020/hash/1f47cef5e38c952f94c5d61726027439-Abstract.html
at the very least, this work should be mentioned (the above arxiv links are published at ICLR 20 & 21 to the best of my knowledge)

l. 90: no, at least not in contemporary neural probabilistic forecasting models. A lot of work is concerned with non-IID errors and non-normal errors (see pointers to literature in the rest of this review; and apart from these there is quite some work on using normalizing flows or GANs for complex forecasting distributions).

in the related work, I would have expected a discussion of deep state space models which should address many of the issues, e.g.,
https://papers.nips.cc/paper/2018/hash/5cf68969fb67aa6082363a6d4e6468e2-Abstract.html
https://proceedings.neurips.cc/paper/2020/hash/afb0b97df87090596ae7c503f60bb23f-Abstract.html and references therein. How would the techniques that the authors present here extend to these cases?

in the empirical comparisons, I would have expected to see some popular neural forecasting models. NBEATS is one such model that is missing. To the best of my knowledge implementations of NBEATS and DeepAR and other models exist in open source. A vanilla LSTM is, in my opinion, a less useful baseline for the empirical comparisons.

The authors limit themselves to linear, first-order auto-correlated errors. It's not immediately clear to me why such a strong restriction would yield such high benefits and why these simple error structures couldn't be compensated for via other approaches, such as increasing the capacity of the neural networks. Also, it would be interesting to discuss the benefits of a more complex auto-correlation structure since we are already in a neural network context. E.g., couldn't equation (4) itself be a neural network whose parameters we could attempt to learn?



**Time Spent Reviewing:**

6h

---

> ### Author Response · Authors · 2021-08-09
> **Response to Reviewer vjG1**
>
> We thank the reviewer for their thorough review, comments, and constructive feedback. We agree with the reviewer “that taking auto-correlated errors into account is an important topic that hasn't received the attention it deserves in the ML literature and the authors build a convincing case for their technique.” We hope the reviewer might consider increasing their rating on this basis so that the present work can reach the ML community, with the aspects raised by the reviewer highlighted as good opportunities for future work and extensions. In that spirit, we provide answers to the comments below.
>
> Q1: Question about line 85.
>
> A1:
> First, we agree that it is our mistake to write “MSE is equivalent to MLE” on line 85. We will move that statement and instead put it after line 90, after assuming that the errors are normally distributed.
> Second, DeepAR only uses negative-binomial likelihood for count data but still uses Gaussian likelihood for real-valued data. In our data, although some series in a multivariate time series dataset can be count data, we treat all series as real-valued data because none of the datasets consist entirely of count data. In general, for real-valued data, assuming Gaussianity remains the most common practice in many machine learning/econometrics applications. However, we would like to point out that our method is still applicable even if the error terms are not distributed normally. As long as all error terms have the same distribution and follow Equation (4), our method can still adjust for the autocorrelated errors, just that instead of MSE, a different loss function would be used.
> The reviewer question about how our approach extends to non-MSE based approaches is a good area for future research. We would like to emphasize that although lifting the Gaussianity assumption is an important future research direction, research about time series forecasting under the Gaussianity assumption is also important. Many state-of-the-art works after 2019 continue to focus on normally distributed errors (i.e., using MSE). For example, [30, 48] cited in the paper. We think it is important that the community be aware of the impact of auto-correlated errors for these widely assumed Gaussian cases, and benefit from an effective method to mitigate them.
> Finally, it is not correct to say that the authors are unfamiliar with the state-of-the-art in probabilistic neural forecasting. Indeed, the authors cite DeepAR ([41] in the paper; we will fix the lowercased “AR” in the reference because latex automatically turns every non-first character into lowercase). We cited DeepAR but not the other two because DeepAR is the earliest one; we instead cited “Neural forecasting: Introduction and literature overview” ([5] in the paper) for a broader reference to other neural forecasting papers. We do agree that we can better highlight the related literature to more clearly position and scope the current paper.
>
> Q2: Question about line 88.
>
> A2:
> Extension to probabilistic forecasting is definitely a future direction. We will add sentences talking about probabilistic forecasting, and cite relevant papers as a future direction in the Conclusion and Broader Impacts section.
> However, we would like to emphasize that our method can be applied with quantile regression, which is a simplified form of probabilistic forecasting to estimate the confidence interval. If one views the series corresponding to the target quantile as a new time series, NNs that forecast the new time series should also adjust for autocorrelated errors and our method can be applied. We can add a note on that point to the final version of the paper.
> From a broader view, our method is applicable to many tasks involving time series data, including regression, classification and anomaly detection. Within each task, there are many possible formulations, so in this first paper we can only focus on experimenting with one formulation for each task. For the task of forecasting, we choose point estimate forecasting instead of probabilistic forecasting in part because it is the most natural formulation to test the effectiveness of our method on time series forecasting. But again, probabilistic forecasting is definitely an important extension that should be discussed in the future.
>
> Q3: Question about line 90.
>
> A3:
> We agree that many contemporary neural probabilistic forecasting models do not assume IID normal error terms, but in general, this is often the assumption, especially in many papers proposing new NN architectures for forecasting. That is why we use the term “Usually” to start line 90.
> We didn’t discuss thoroughly multiple different formulations for forecasting, for two reasons as discussed in A2. First, we want to show that our method is effective on various tasks and not just forecasting, so we can only focus on one formulation of forecasting. Second, our method can be directly applied with the formulation we present in the paper, which is a simple and popular formulation. This is the most natural formulation to show the effectiveness of our method on forecasting. Extending to other formulations, such as state space models or probabilistic forecasting, is definitely a good suggestion and a future direction.
>
> Q4: In the empirical comparisons, I would have expected to see some popular neural forecasting models, such as N-BEATS and DeepAR.
>
> A4:
> We are not sure what models are considered popular and what models are not. However, as mentioned in the paper, we chose the set of models to include multiple RNN, CNN, GNN, and Transformer based models, which covers the four fundamental building blocks of NNs. Importantly, the model needs to be designed for multivariate time series input/output. N-BEATS is designed for univariate forecasting. DeepAR is designed for multivariate forecasting but the model is trained on each feature series separately. We can add mention in the paper about those limitations to N-BEATS and DeepAR as reasons for not including them in our comparisons. Although we didn’t explicitly test our method on a univariate time series dataset, we tested on many datasets and some of the datasets consist of only about 10-feature series.
>
> Q5: It's not immediately clear to me why such a strong restriction would yield such high benefits.
>
> A5:
> Quantifying the reasonable amount of benefits is not trivial. For example, if we assume all errors come from irreducible errors which are just random noise, then we would expect the resulting MSE to be equal to the variance of the noise. However, if the noise is autocorrelated with coefficient $\rho$ and we don’t adjust for it, the (observed) variance of the noise becomes $1/(1-\rho^2)$ as derived in Appendix A. Now, if $\rho=0.2$, the variance is 4% larger than the original one, which implies that adjusting for autocorrelated errors improves the performance by 4%. In practice, there is apparently quite often some amount of reducible noise, and thus the benefits shown in the experiments are reasonable. However, the example and discussion above shows that quantifying a generally expected amount of benefit is not trivial, because it depends strongly on the source of the errors.
>
> Q6: Why these simple error structures couldn't be compensated for via other approaches, such as increasing the capacity of the neural networks.
>
> A6:
> Increasing the capacity of the neural networks indeed decreases the need for adjustment because it at least decreases the degree of model misspecification, which is empirically verified in Section 5.8.
> Other approaches might also exist that can well compensate for the autocorrelation errors. For example, as mentioned in Section 5.9 Limitation, a better model architecture will make our method less useful. However, our paper also provides empirical critical values to act as a guideline to determine whether adjusting for autocorrelated errors is necessary or not.
>
> Q7: Also, it would be interesting to discuss the benefits of a more complex auto-correlation structure since we are already in a neural network context.
>
> A7:
> It is possible that a more complex structure would be beneficial, but we didn’t discuss it because (1) a linear model is a good choice for simple and basic modeling, (2) we follow the work in the field of econometrics, where most of the discussions about autocorrelated errors happen, and most importantly (3) if we use a NN for modeling the errors (i.e., replacing $X_t - \rho X_{t-1}$ with $X_t - f(X_{t-1}; \theta)$), the overall model becomes deeper with many more parameters, then it becomes unclear whether the improvement comes from adjustment or from a bigger model. In summary, we think for this first paper that a linear structure is a good choice for modeling errors without increasing the overall model complexity too much. But future work on more complex auto-correlation structures is an interesting avenue.

---

> > ### Comment · Reviewer_vjG1 · 2021-08-20
> > **thanks for the rebuttal**
> >
> > Thank you for having addressed some of my questions. I'm happy to raise my score a little bit, but concerns remain.
> >
> > For the comment on Q1:
> > - no, the likelihood in DeepAR can be chosen flexibly. To the best of my knowledge, the default likelihood for DeepAR is actually not the Gaussian but the student-t density. For example in the M5 competition, some authors proposed the Tweedie distribution: https://www.sciencedirect.com/science/article/pii/S0169207021001151 I acknowledge that this article appeared after the submission of the present article, but the point is that this is well known and more so, available in code, e.g., in PyTorchTS or GluonTS. Much of the work in modern forecasting methods is in having flexible likelihoods (normalizing flows, GANs, energy-based models, structured likelihoods, see articles by K. Rasul or S. Rangapuram). In summary, my doubts on remain.
> >
> > comment on Q4:
> > Point take on the popularity of forecasting methods. I should have said that there are dedicated forecasting models available that clearly don't have the problems that this paper tries to address. However, more naive, ad-hoc models would have these problems, which I acknowledge.

---

> > > ### Author Response · Authors · 2021-08-22
> > > **Response to Reviewer vjG1**
> > >
> > > Thank you very much for raising the score. We very much appreciate your recognition. Here, we provide additional responses to your latest concerns.
> > >
> > > Response to Q1:
> > >
> > > In general, yes, we knew that likelihood in DeepAR is flexible. But the DeepAR we refer to in the original rebuttal is the one in the original paper (https://www.sciencedirect.com/science/article/pii/S0169207019301888), which uses negative-binomial likelihood for count data and Gaussian likelihood for real-valued data. We apologize for the misunderstanding.
> > >
> > > To the best of our knowledge, the original DeepAR doesn’t use the student’s t-distribution. We are not sure, but we only know that the student’s t-distribution is set as the default distribution in the GluonTS implementation of DeepAR.
> > >
> > > We completely agree with you that many modern forecasting methods propose flexible likelihoods that might or might not have the problem of autocorrelated errors. However, we would like to emphasize that Gaussianity remains the most single most common likelihood in many machine learning/econometrics applications. Our main contributions are pointing out the issue of Gaussianity in time series data and proposing a method to resolve the issue in neural networks. Extending to other likelihoods is definitely a future direction.
> > >
> > > Response to Q4:
> > >
> > > Yes, we agree that more complicated, flexible models might not have the issue of autocorrelated errors. However, our paper provides a possible explanation for the improvement of those complicated models over the naive ones. In other words, part of the improvement of those complicated models might come from implicitly adjusting for autocorrelated errors. From this point of view, our paper is a step toward understanding and explaining the advantages of those complicated models.

---

### Decision · Program_Chairs · 2021-09-27

**Decision:**

Accept (Poster)

**Comment:**

All reviewers are overall positive, the paper addresses a problem that is important, and the authors have provided thoughtful and useful replies to the reviewers.

The paper proposes an elegant way to deal with autocorrelated errors in time series usable with a variety of existing neural nets. The idea is to learn the net’s parameters jointly with an autocorrelation coefficient parameter. The paper combines insights from how autocorrelation error is dealt with in econometrics with deep learning for time series analysis. The method may be a major advance, since it can improve many existing neural time series models.